# Experimental Study on the Performance Decay of Thermal Insulation and Related Influence on Heating Energy Consumption in Buildings

**Diana D'Agostino** [1] , **Roberto Landolfi** [2] , **Maurizio Nicolella** [2,*] **and Francesco Minichiello** [1]

1   Department of Industrial Engineering, University of Naples Federico II, 80125 Naples, Italy; diana.dagostino@unina.it (D.D.); minichie@unina.it (F.M.)
2   Department of Civil, Architectural and Environmental Engineering, University of Naples Federico II, 80125 Naples, Italy; roberto.landolfi@unina.it
*   Correspondence: maurizio.nicolella@unina.it

**Abstract:** The External Thermal Insulation Composite System (ETICS) is one of the most common passive strategies to obtain energy savings in existing buildings. Despite previous studies dealing with ETICS durability in real building case studies or involving accelerated ageing tests in climatic chambers, little progress has been made in the knowledge of the long-term durability and energy efficiency of the system. In this work, following previous experiments on ageing cycles, different climatic chambers are used to accelerate performance decay by simulating the natural outdoor exposure to assess the thermal transmittance decay of a building wall. After evaluating through laboratory tests the decay of the thermal performances of grey expanded polystyrene sintered (EPS) and polyurethane, the results are applied to an existing building. The case study building is virtually located in three different Italian climatic zones and an evaluation in terms of thermal transmittance values and their influence on heating energy consumption is made by using the dynamic simulation software DesignBuilder. The results show no significant variations during ETICS service life; the thermal performances are reduced little over time and therefore there is an increase in consumption for building heating of about only 2% after a time t1 equal to 8 years.

**Keywords:** External Thermal Insulation Composite Systems (ETICS); accelerated ageing test; ageing; building; dynamic simulation; thermal resistance; energy savings

## 1. Introduction and State of the Art

Nowadays, for a sustainable human development, the greatest challenge has become the reduction in the energy consumption and environmental impacts of any productive activity. In recent years, buildings and the construction sector accounted for 36% of final energy consumption and 39% of energy and process-related carbon dioxide ($CO_2$) emissions [1]. Moreover, around 77% of the global final energy demand in buildings is for heating and cooling end-uses, including space heating and cooling, domestic hot water, and cooking, while the remaining final energy demand in buildings (23%) is for electrical end-uses, including lighting and appliances [2]. The European Directive (EU) 2018/2002 on energy efficiency establishes the headline energy efficiency target by 2050 [3].

The building energy retrofit is one of the most important goals to obtain an energy-saving long-term scenario, which has a lower environmental impact compared to demolition and new construction [4,5]. Furthermore, this type of intervention can prolong the useful life of the building, and the market value of the construction increases [6]. A retrofit intervention can highly reduce the energy consumption in buildings mainly if it contemplates the use of passive energy saving strategies [7], as indicated in the European Directive 2010/31/EU [8]. Pajek and Kosir [9] highlight the contribution of passive strategies on

heating and cooling energy consumption in the retrofit intervention of a residential building for different European climatic conditions. The highest reduction in yearly energy consumption (up to 30 kWh/m$^2$) is due to passive strategies.

Building energy consumption is mainly due to the high heat transfer through the opaque building envelope. From this point of view, the most efficient passive solution is the thermal insulation of the opaque components [10,11], which can be easily applied to both existing buildings and new ones. Different insulation materials and their thicknesses represent variables that strongly influence the building thermal behavior. Neya et al. [12] highlight how the energy performance of walls can be modified by a proper choice of several materials to also consider the thermal comfort of the users. Furthermore, by analyzing glass wool and straw mixed with lime, the same thermal performance can be obtained by varying the thickness. However, D'Agostino et al. [10] highlight how, when increasing the thickness of the insulation layer to improve winter thermal conditions, after a certain thickness (8–10 cm), this increase can worsen the building's energy performance in summer. In Geng et al. [13], the passive techniques are coupled with a structural panel to easily improve the thermal conditions at the same time as a structural retrofit intervention, to improve the cost–benefit ratio. Huang et al. in [14] show how the optimal insulation thickness is a key point to evaluate energy-saving effects and economic benefits. They find that the minimum optimum thickness of aerogel insulation material is 3.7 mm, allowing an energy saving in heating consumption of about 18.2%. Several papers propose optimized models on the optimum insulation thickness of walls and roofs of existing buildings based on different parameters, such as the primary energy consumption, global costs or pollutant emissions [15,16].

The External Thermal Insulation Composite System (ETICS) is one of the best solutions to obtain an energy efficient building and widely used for new buildings, but mainly for the energy improvement of existing buildings, due to the little invasiveness of the intervention compared to others. This retrofit strategy significantly improves the thermal performance of façades. The insulation materials are various, and each one has not only different technological characteristics, but above all, various characteristics that can lead to different aging processes. Recently, the scientific literature has focused on the durability of building components during their lifetime. Extending the durability of building parts allows for a reduction in maintenance costs and even renovation costs, decreasing the environmental impact and promoting sustainability. From this perspective, ETICS is often used for its well-known advantages, such as the decrease in global thermal loads (mainly heating loads), increase in the thermal inertia of the building envelope, the elimination of thermal bridge effects, and the protection of the envelope from summer and winter thermal shocks, improving building service life. This technology, created in 1960 by E. Horbach in Germany [17], is indicated along with other names: Exterior Insulation Finish Systems (EIFS) in Canada and U.S.A., and External Wall Insulation Systems in Ireland and United Kingdom. ETICS are a kit under the Construction Products Regulation (CPR) [18], consisting of prefabricated components applied directly to the façade onsite. According to UNI/TR 11715 [19] and EAD 040083 [20] shared terminology, these components perfectly fit together thanks to their system holder tested design.

Although the scientific literature commonly evaluates building components service life by means of performance decay assessments, for ETICS, this approach has only been followed for a few years. Some studies investigated ETICS durability focusing on the long-term analyses of naturally aged samples accounting existing building case studies. Amaro et al. carried out a statistical survey of the pathology, diagnosis and rehabilitation of ETICS in walls due to inspection of 146 façades, demonstrating that anomalies in ETICS can be prevented by the proper design, application, and choice of appropriate materials. Especially breakage anomalies and façade flatness anomalies can be reduced, demonstrating the importance of design and maintenance stages for ETICS service life [21]. These findings are comparable to Swedish research consisting of a survey of 821 buildings, which shows that moisture, resulting from the poor connections of windows and doors, is pri-

marily responsible for the decay of joints and fixing devices—wetting the materials inside the stud wall and causing mold growth [22]. A similar statistical approach was carried out in Norway analyzing 61 buildings with ETICS cladding where a total of 150 causes of defects were recorded: defects associated with flashings against precipitation, incorrect reinforcement mesh, insufficient thickness of render, a default render mix in undesirable setting conditions, shrinkage and temperature movements in the render, incorrect end laps against adjoining structures, faulty anchorage of the system, microorganism growth in/on the render, variations in render thickness over the insulation boards, vibration movements in the substructure, settling, incorrect choice of paint or incorrect cleaning before painting, insufficient impact resistance, lack of pigment, and mold growth behind the ETICS [23]. To assess the thermal–hygrometric performance and the state of conservation of the exterior envelope, Stazi et al. studied the thermal performance verifying the efficacy of an ETICS intervention after 20 years and showing durability from the thermal–hygrometric and mechanical point of view of the external insulation applied in the 1980s [24]. The previous studies highlighted the importance of a correct choice, maintenance plan, design and application of materials. Otherwise, other studies dealt with short-term investigations of accelerated ageing in climatic chambers, as it is well known that accelerated ageing has a great advantage due to the possibility of obtaining results quickly, compared to exposure to natural atmospheric agents. Recently, many researchers have found satisfactory long-term ETICS durability by analyzing building products subjected to appropriate accelerated ageing in laboratory. The major difficulties are attributable to the assessment of the proportion between natural exposure and accelerated ageing cycles.

Researchers focused on the evolution of decay in ETICS adopted a common methodology to design an ageing cycle for ETICS: the analysis of weather data of the locality and assessment of which climatic agents had to be included, and their intensity and frequency reproduced in laboratory tests within a relatively short timeframe compared to the natural ageing of the outdoor climate. These studies were conducted by some Finnish researchers who provided examples of ageing methods, of climate ageing laboratory equipment, and of building-product properties to be tested before, during, and after ageing [25], to evaluate a new calculation method to estimate accelerated ageing related to natural outdoor climate [26]; materials and components used for building envelopes were exposed to UV light, heat radiation, water, and frost during the testing of new building solutions. Bochen assessed the behavior over time of external mineral plasters and then of ETICS external render measuring the change of porosity [27–29] after an accelerated ageing test involving UV, heat and cold cycles, and freeze and thaw cycles reproduced by a rotational method within different climatic chambers, each of which reproduced different agents. The Northest NT Build 495:2000 [30] uses a rotation chamber for the accelerated ageing simulator. This method was carried out in Northern Europe in most of the studies [23–26] and consists of an apparatus in which specimens can be rotated within four different climate zones.

In Italy, researchers have focused on the evolution of decay in ETICS and on the correlation between natural exposure and accelerated ageing cycles [31]. Thanks to the analysis of weather data in Milan, it was possible to establish which climatic agents had to be included, and their intensity and frequency [32]. This phase, leading to design ageing data cycles, produced important findings to evaluate the reference service life by using the analysis of the reference climatic conditions [33]. Moreover, a comparison between degradation occurred with artificial accelerated ageing and degradation occurred with short-term outdoor exposure was possible, also obtaining a useful ratio (the so-called "re-scaling factor") [18] basing on the ISO 15686-2 standard [34]. This aforementioned methodology is one of the most complete and valid because it allows the assessment of performances, such as time shift and decrement factor, by means of the specimen used as a door of the climatic chamber, achieving a detailed characterization of thermal performance. Different approaches define a generic North Atlantic context without considering the specific climatic context, basing on the ETAG004 [35] hypothesis. The method was used to design a short-term laboratory accelerated ageing test by means of cycles as follows: UV

cycle, winter cycles involving rain, freeze and thaw and summer cycle involving dry heat and rain [36].

The first studies within naturally aged ETICS showed that considerable anomalies were noticed even only a few years after their application, and this raised doubts about their long-term durability. Different results, obtained throughout accelerated ageing cycles, showed that ETICS performances have minor variations.

Not all studies agree on this finding. For example, Parracha et al. studied the effects of one-year natural ageing and artificial ageing of three ETICSs [37]. They highlighted a relevant loss of surface hydrophobicity after ageing: in 24 h, the naturally aged systems absorbed up to 73% of water, whereas the artificially aged ones absorbed up to 432%. Drying resistance increased up to 122% after artificial ageing. The artificially aged systems were the only which showed traces of mold growth. Aesthetic alteration was confirmed by the color shift for all systems after ageing.

To the best of the authors' knowledge, international research is focused on energy savings achieved through passive techniques, such as thermal insulation, without focusing on how the performance of the building envelope changes when the insulating layer undergoes aging processes. Furthermore, several studies focused more on surface degradation than on thermal performance and the few that studied thermal resistance decay were not able to estimate its influence on building heating energy consumption. The aim of the paper is to experimentally evaluate through laboratory tests how the thermal performance of ETICS (based on an insulating layer of grey EPS or polyurethane) changes when subjected to aging processes. The experimental results are successively applied to a case study building virtually located in three different Italian climatic zones to evaluate how the aging of the insulation affects the thermal performance of the wall and consequently the energy consumption for the building heating in winter. Several comparisons are made in terms of variation of the thermal conductivity of the material, of the thermal transmittance of the wall composed of tuff masonry and insulating layer, and of the energy consumption for winter heating. An evaluation in terms of the worsening of thermal transmittance and its influence on heating energy consumptions is made by using the dynamic simulation software DesignBuilder. The main innovative aspect of this paper is the linking of the experimental results on the decay of thermal performances of ETICS with the evaluation of the related energy consumption for the building heating.

## 2. Materials, Methods and Experimental Investigation

The first step of the study consisted of the exposure of a sample of ETICS on a masonry wall to various accelerated laboratory ageing cycles. In order to evaluate thermal performances of ETICS, before and after ageing, degradation was assessed by decay evaluation of the thermal transmittance of the wall. The methodology of the research is relevant because it allows the assessment of the last parameter by measuring the thermal transmittance of an insulation panel used as a door of a climatic chamber, achieving a detailed characterization of thermal performance. By using this method, the sample divides the internal part of the chamber, which simulates the outdoor exposure, from the outside of the climatic chamber, which is the indoor of the building and then constitutes the indoor environment.

The components of the analyzed ETICS were:

1. Adhesive;
2. Insulation products;
3. Mechanical-fixing devices;
4. Rendering systems, typically consisting of a base coat, reinforcement (glass fiber), and a finishing coat/decorative coat;
5. Secondary materials (any supplementary component/product used to form joints or to achieve continuity).

The laboratory test used specimens with the same materials as components 1, 3 and 4, while component 2 (insulating material) was changed. A white cement adhesive was used as component 1, PVC fixing devices as component 3, and, as component 4, a cement base coat, a

reinforced glass fiber and a thick coat. Indeed, component 2, the insulating material, varied among the most used ones for ETICS. This aspect is better analyzed in Sections 2.1 and 2.2.

### 2.1. Insulating Materials Used in the Experimental Investigation

The experimental part of the study expanded the aim of other research works ([26,27,30,36,37]), and provided an evaluation of the development of façade decay over time, how it affects the thermal performances and thus the building heating energy consumption.

The reference for investigating all the insulating materials for ETICS was UNI/TR 11715 [19], a technical standard that includes a complete list of all the thermal insulation products, i.e., cellular glass, mineral wools, such as rock wool and glass wool, expanded polystyrene, extruded expanded polystyrene, polyurethane, wood fiberboard, cork and polyester fiberfill.

This research investigated two of the most widely used and well-performing materials for vertical envelopes, i.e., polyurethane (PU) and grey expanded polystyrene (EPS), applied in the case study of a single-family building virtually located in three different Italian climatic zones (Palermo, Naples and Turin).

### 2.2. Specimens' Characteristics

Specimens were prepared according to EAD 040083 [20]. Considering that the main objective of the research was to provide a relative comparison of the energy consumption before ageing and after ageing, two samples were created by keeping the same stratigraphy, while varying the insulating material between the grey expanded polystyrene and polyurethane. Therefore, two kinds of samples were packaged for each insulating material (Grey EPS, PU) using the same ones for the other layers according to the related ETA (European Technical Assessment).

Each specimen was realized with the following characteristics:

- Support of wooden OSB (Oriented Strand Board) (thickness = 2 mm) and outdoor plasterboard panel (thickness = 12.5 mm) instead of masonry wall, because not only the aim of the research was focused on the interaction between ETICS and environmental loads, and not on the back support, which has a negligible influence on durability of the system, but also as the insulation is supposed to be placed outside the support (external insulation).
- Skim-coating adhesive, a mineral adhesive/skim coat in powder form made of unsaponifiable resins, high-resistance Portland cement and selected sands with a maximum particle size of 0.6 mm.
- Thermal insulating material, with different widths as a consequence of the different conductivity, as specified in Table 1, in which the system name refers to the commercial name of the company that provided materials; the target value of thermal transmittance was chosen in compliance with Italian law requirements (D.M. 26 June 2015 [38]), considering the hypothesis of a busser double brick wall with internal cavity.
- Base coat with embedded reinforcing fiberglass mesh: the base coat material is the same skim-coating adhesive.
- Finishing coat: coating based on acrylic resins in dispersion within additives that facilitate the application and formation of a film as well as marble granules and quartz sand with controlled absorption; max. particle size of 1.2 mm.

The insulating material was covered by the whole rendering system (base coat, glass fiber, and finishing coat) because the glass fiber mesh was turned round the edges of the specimen covering five faces of the insulating panel.

IVAS SpA, an international industry leader operating in building finishes, offering products, solutions, systems, integrated technologies in construction market, provided all materials and accessories according to its certified systems within the main insulating materials producers.

**Table 1.** Thermal characteristics of different samples with two types of thermal insulating materials.

| Insulating Materials | Commercial ETICS Name | Insulating Panel [1] | | Whole ETICS | |
|---|---|---|---|---|---|
| | | s (mm) | $\lambda_D$ (W/mK) | Thermal Transmittance (W/m$^2$K) | Thermal Resistance (m$^2$K/W) |
| Polyurethane | Termok8 Slim | 50 | 0.028 | 0.51 | 1.97 |
| Grey EPS | Termok8 Modulare Biostone | 60 | 0.031 | 0.47 | 2.12 |

[1] s = thickness (mm); $\lambda_D$ = declared conductivity.

Unlike other previous studies, big samples were used measuring 55 cm × 60 cm because they were used as door of a climatic chamber.

Specimens were built with 3 insulation panels (of the same material) in order to reproduce one T joint in the middle of the sample where it was placed a mechanical fixing device (Figure 1), because the scientific literature and common professional experience show damages especially in the correspondence of T joints. Then, as shown in Figure 1, the whole specimen was mechanically fixed and bounded and then it was cured for 28 days before testing.

### 2.3. Experimental Approach by Means of Accelerated Ageing

The system is naturally stressed by critical agents and ageing factors. The aim of this phase was ageing the aforementioned samples by means of accelerated ageing cycles capable of reproducing:

- Summer and winter thermal shocks;
- Freeze–thaw cycles;
- Driving rain;
- Cyclic variations in temperature and relative humidity.

Therefore, according to a previous study [36], it was found that the relevant agents for this building component are these of the aforementioned list and it was shown that, in short ageing process, it is possible to take into account only the frequency of events over a critical threshold. Then, according to previous proceedings of the experimental process [39], based on the ISO 15686-2 code [34], it is possible to weigh critical events comparing natural ageing with the accelerated one. By the use of the EAD 040083 model [20], the real behavior and stress of ETICS under defined conditions were simulated.

The European Assessment Documents (EAD) are useful to obtain a European Technical Assessment (ETA) by a designated technical assessment body (TAB) affixing the CE marking. ETICS manufacturers starts the evaluation process requiring an ETA, which is based on reference EAD. The reference EAD for ETICS is the EAD 040083 [26], a technical guide for ETICS property assessment based on the previous ETAG 004 [21].

In EAD 040083, through a pre-established test program, the components, organized in a specific thermal insulation system, are tested in order to verify all the main characteristics and performances.

EAD 040083 has already analyzed ETICS decay factors and mechanisms, in order to obtain an ageing cycle suitable for a generic climatic context. This study adopted the EAD 040083 method because it designs ageing cycles similar to the EAD and ETAG ones, and, considering the purpose of investigating ETICS decay according to annex D.3 of EAD 040083, proposes the combined hygrothermal and freeze-and-thaw cycles test. These conditions are widespread across Europe.

This research undertook this EAD methodology with minor variations in order to reproduce real outdoor exposure during years.

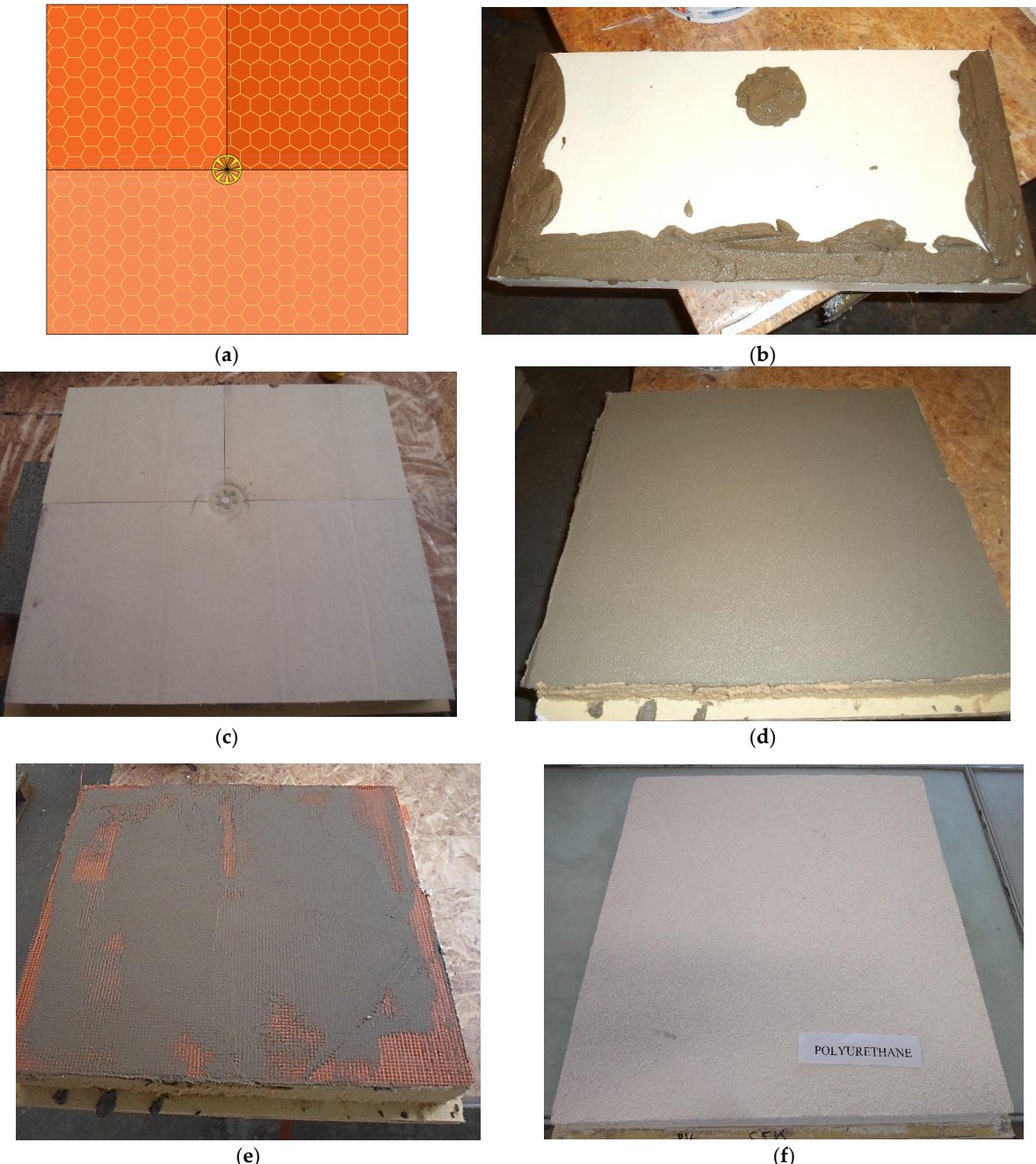

**Figure 1.** (**a**) Designed sample with 3 different thermal insulating panels and a T joint where a mechanical fixing is usually placed; (**b**) Bonding of insulation panels cut and turned; (**c**) The sample within insulating panels just bonded to plasterboard panel; (**d**) The sample, with the addition of the first plaster base coat with embedded reinforcing fiberglass mesh; (**e**) The sample, with the addition of the second coat with embedded reinforcing fiberglass mesh; (**f**) The sample, constructed and finished with the top coat.

However, EAD targets require only correspondence with the basic requirements of the European regulations and directives, including the Construction Products Regulation (CPR). Then, although the aforementioned EAD standard methodology perfectly fits with ETICS behavior assessment during lifetime, EAD targets limited it. To achieve detailed characterization of ETICS durability, further observations were needed. Since the study was concerned with slowly ageing materials to evaluate ETICS behavior during service life comparatively, in reasonable time, the research set itself apart from the EAD approach

because it obtained a longer-term investigation target in accelerated ageing tests, expanding the aforementioned EAD cycles to cycles that exceed EAD limits. This approach was used because the study was concerned with slowly ageing materials to evaluate ETICS behavior during service life comparatively, in reasonable time, in line with the previsions of EAD 040083.

This research can extend our understanding on ETICS service life and, overall, on the influence of the ageing of the insulation on the winter energy performance of a building, based on the experimental data of laboratory ageing tests.

### 2.4. Designed Accelerated Ageing Cycles

According to EAD, the cycles were designed as follows:

- A total of 80 heat–rain cycles: each cycle took 6 h and consisted firstly in heating up to 80 °C (rising for 1 h) and maintaining the temperature at (80 + 5) °C for 2 h (total of 3 h), then spraying for 1 h with $1.5 \pm 0.5$ l/m$^2$ min amount of water and water temperature at $15 \pm 5$ °C, and thirdly leaving for 2 h for drainage at $20 \pm 5$ °C;
- A total of 7 heat–Cold cycles: each cycle lasted 24 h, comprising an initial exposure of 8 h to $-10 \pm 2$ °C (fall for 2 h), then 9 h to $70 \pm 2$ °C (rise for 1 h) and maximum 30% RH and finally an exposure of 7 h to $-10 \pm 2$ °C (fall for 2 h);
- A total of 15 freeze and thaw cycles: each cycle lasts 24 h, comprising an initial exposure for 8 h to water at $23 \pm 4$ °C by immersion of the specimens, with the skin submerged in a water bath, according to the method described in EAD 040083 (Section 2.2.7) [26], then, freezing to $-20 \pm 5$ °C for 14 h, and a final insertion in the stove at +50 °C for 2 h.

Table 2 indicates the main parameters of the accelerated ageing cycles.

**Table 2.** Designed accelerated ageing cycles: the main parameters are number of cycles, machine/phase, time interval, temperature and relative humidity conditions.

| Cycle Typology | Cycle Number | Phase/Insertion in: | Phase Dt (h) | Cycle Dt (h) | T (°C) | RH (%) |
|---|---|---|---|---|---|---|
| Heat–Rain | 80 | Heater | 3 | | 80 | 50 |
| | | Spryer | 1 | 6 | $15 \pm 5$ | 100 |
| | | Drainage | 2 | | $15 \pm 5$ | 100–50 |
| Heat–Cold | 7 | Climatic chamber | 8 | | 70 | 30 |
| | | Climatic chamber | 16 | 24 | $-10$ | 0 |
| Freeze and thaw | 15 | Water bath | 8 | | $15 \pm 5$ | 100 |
| | | Freezer | 14 | 24 | $-20$ | 0 |
| | | Stove | 2 | | 50 | 50 |

### 2.5. Performance Decay Assessment

Compared with other studies, the proposed methodology for decay assessment took into account thermal performance. Basing on the previous literature concerned with ETICS decay analysis, this study aimed to analyze the most important performance, i.e., thermal resistance.

Considering ETICS were created to improve building thermal performance and energy efficiency, thermal resistance or thermal transmittance is the only performance indicator to evaluate the thermal requirement.

Thermal resistance was determined by means of guarded hot plate and heat flow meter method according to the EN 12667 [40]. The whole sample was placed inside the standard equipment with flat and parallel sides (Figure 2).

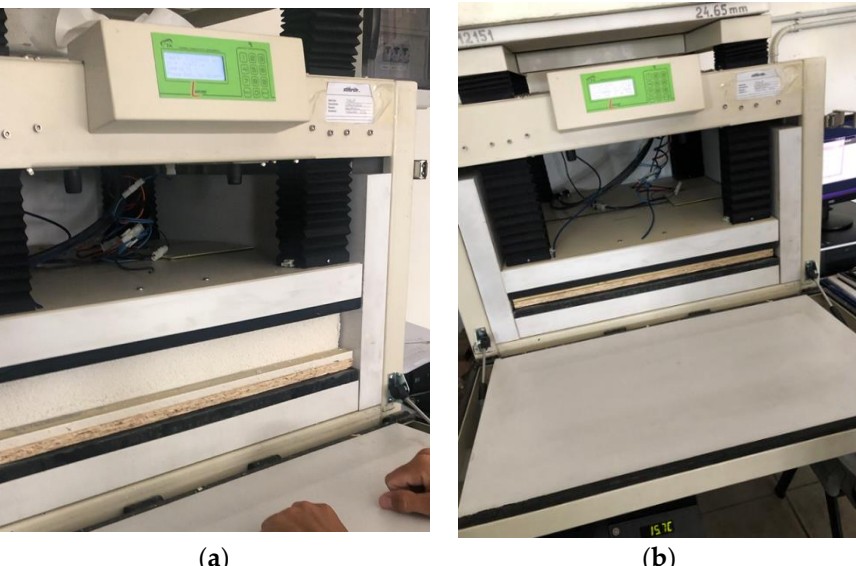

(**a**)          (**b**)

**Figure 2.** (**a**) Thermal resistance measurement according to EN 12667; (**b**) Thermal resistance measurement of the stratigraphy needed for the determination of the thermal resistance of each single material.

When investigating the performance of thermal insulating materials, the main reference parameter is thermal conductivity, which is inversely proportional to the thermal resistance according to the following Formula (1):

$$R = \frac{1}{h_i} + \Sigma \frac{s_n}{\lambda_n} + \frac{1}{h_e} \tag{1}$$

The adopted procedure (EN 12667) allows the evaluation of the thermal resistance, from which the conductivity of single materials of the stratigraphy was calculated (the conductivity of the materials other than insulation is measured). The internal (hi) and external (he) heat transfer coefficients were assumed to be equal to 7.7 W/m²K and 25 W/m²K, respectively. These values are often used in heat transmission analyses to evaluate the thermal transmittance of the vertical walls of buildings.

The experiment followed these phases:

- Realization of ETICS samples by varying the insulating material between grey EPS and PU without varying the other layers;
- Measurement of decay before the test; collection of data related to sampled ETICS thermal resistance at time T0;
- Execution of accelerated ageing test on the samples in climatic chambers;
- Measurement of the state of decay after accelerated ageing cycles; collection of data related to sampled ETICS thermal resistance at time T1;
- Realization of the performance–time curve for each of the solutions represented by the two samples.

The first measurement concerned the thermal transmittance, according to the EN 12667 [40], as described in Chapter 3.1. This allowed to verify the real thermal transmittance, different from the calculated one, which derives from the value of thermal conductivity shown in Table 1.

Table 3 reports:

- The declared conductivity and related calculated thermal transmittance and thermal resistance of whole ETICS;
- The experimentally measured thermal transmittance and thermal resistance of the studied ETICS before accelerated ageing cycles, at time T0;
- The experimentally measured thermal transmittance and thermal resistance of the studied ETICS after accelerated ageing cycles, at time T1.

**Table 3.** Characteristics of samples with different types of thermal insulating materials, calculated, then measured before accelerated ageing cycles and finally measured after the accelerated ageing cycles.

| Symbol | Thermal Characteristics | Whole ETICS within following Insulating Materials | | Measurement Unit |
|---|---|---|---|---|
| | | Polyurethane | EPS | |
| $\lambda_D$ | Declared thermal conductivity | 0.028 | 0.031 | W/mK |
| $U_C$ | Calculated thermal transmittance | 0.51 | 0.47 | W/m²K |
| $R_C$ | Calculated thermal resistance | 1.97 | 2.12 | m²K/W |
| $\lambda_0$ | Measured thermal conductivity before ageing | 0.025 | 0.034 | W/mK |
| $U_0$ | Measured thermal transmittance before ageing | 0.44 | 0.49 | W/m²K |
| $R_0$ | Measured thermal resistance before ageing | 2.28 | 2.02 | m²K/W |
| $\lambda_1$ | Measured thermal conductivity after ageing | 0.026 | 0.034 | W/mK |
| $U_1$ | Measured thermal transmittance after ageing | 0.46 | 0.50 | W/m²K |
| $R_1$ | Measured thermal resistance after ageing | 2.18 | 2.02 | m²K/W |

Note: $\lambda_D$ = thermal conductivity of the insulating panel declared by the technical datasheet and calculated according to EN12667. Measurement uncertainties for the given data were calculated 5%.

For the sake of completeness, a thermal resistance test was even carried out according to the EN 12667 [40] on the whole stratigraphy, i.e., OSB, plasterboard and finishing coat (total thickness = 30.32 mm) without the thermal insulating material, to determine the thermal transmittance and the thermal resistance, which were, respectively, 3.78 W/m²K and 0.26 m²K/W.

These measurements show (Table 3) a negligible decrease in thermal performance during ageing for EPS and a very slight decrease for polyurethane. In particular, polyurethane shows a percentage increase in conductivity by only 4%, which can be related to the range of tolerance of the measurement methodology, realized according to EN 12667 standard; however, it is lower than the declared conductivity.

As it is known, in rigid expanded polyurethane foams, thermal conductivity rises in the first years because of the diffusion in the bounding parts of the panel of part of the blowing agents (gases) contained in the closed-cell foam towards the outside [41], partially substituted by air.

In order to assess this phenomenon, technical rules are provided, in the text of the standard EN 13165-Annex C [42], which presents two methods that can be used by manufacturers. During both procedures, the producer has to add to the obtained value a statistical correction factor that ensures an adequate correspondence of the value to that of the declared performance after 25 years. The specific harmonized product standards (Annex A and C of the UNI EN 13165 for polyurethane products) provide the methodologies applied for the determination of the declared thermal conductivity value, $\lambda_D$, which is the average thermal conductivity value of the product for 25 years of service life. This is the reason why PU shows a thermal conductivity measured at time T0 lower than the declared thermal conductivity, as shown in other studies [43,44].

A difference between the declared conductivity of the insulating panels and the measured one can be observed; the explanation can be found in EN 13172 code [45], a code followed by producers of insulating materials, which introduces measurement tolerances.

Moreover, the obtained experimental results show that only for the EPS the declared value is higher than the experimentally evaluated result at time T0; however, on the other hand, EPS shows a greater resistance to aging over time compared to polyurethane.

It is useful to underline that the measurements on thermal resistance of insulating materials were repeated to reduce the measurement errors. No different values were recorded in all measurements, and this ensured that the measurements are reliable.

## 3. Application of Experimental Results to a Case Study

The experimental results were applied on a case study of an existing building to evaluate the influence of the thermal insulation decay on its heating energy performances.

The building is a single-family house (Figure 3), virtually located in three Italian cities with different climatic conditions. The first one is Palermo (Lat. 38°6′43″56 N; Long. 13°20′11″76 E), a city with a very hot climate; the second is Naples (Lat. 40°51′46″80 N; Long. 14°16′36″12 E), a city with mild winters and hot summers; and lastly, Turin (Lat. 45°4′41″16 N; Long. 07°40′33″96 E), a city with a continental climate with very cold winters. The Heating Degree Days (HDD) of Palermo, Naples and Turin are 751, 1034 and 2617, respectively, and based on climatic classification of DPR 412/1993 [46], these cities belong to climatic zone B, C and E, respectively. In the case of climatic zone B, there is an operation period of heating systems from 1 December to 31 March for a maximum of 8 h per day. For climatic zone C, the operation period of heating systems from 15 November to 31 March with a maximum of 10 h per day is mandatory. In the case of climatic zone E, there is an operation period of heating system from 15 October to 15 April with a maximum of 14 h per day.

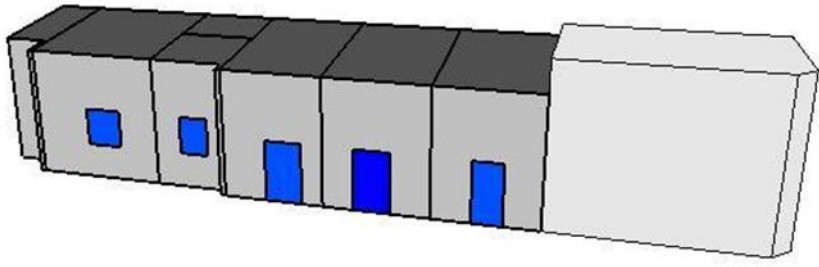

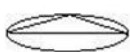

**Figure 3.** Three-dimensional model and orientation of the investigated building.

To apply the results of the experimental tests of decay of ETICS, a comparison between the two different materials was made. The insulation materials were inserted as the thermal coats of a building with yellow tuff masonry walls of two different thicknesses, equal to 60 cm and 35 cm. For each material, the thickness was chosen to guarantee a thermal transmittance value U minor than the minimum values imposed by the current Italian law [30], equal to 0.40 W/m²K, 0.36 W/m²K and 0.28 W/m²K for Palermo, Naples and Turin, respectively.

*Results for the Case Study and Discussion*

The first comparison was conducted with reference to the conductivity λ as declared value, measured values at time T0 and measured value at time T1 equal to about 8 years later from time T0. Thermal transmittance U and building heating thermal load were also calculated in these three different conditions. For the comparison, the chosen thickness of the wall thermal insulation was fixed to a value of 4 cm for Palermo, 6 cm for Naples and 10 cm for Turin. The thermal characteristics of each material for every stage of research and the building heating thermal loads are shown in Tables 4–9. To calculate the heating thermal load, the dynamic simulation software DesignBuilder uses a steady-state method as suggested by ASHRAE (American Society of Heating Refrigerating and Air-Conditioning Engineers) and CIBSE (Chartered Institution of Building Services Engineers).

**Table 4.** Palermo. Thermal characteristics of insulated walls with a polyurethane layer and heating thermal load.

| Polyurethane | | | |
|---|---|---|---|
| **PALERMO** | **Declared** | **Measured t = 0** | **Measured t = 1** |
| Thickness [m] | | 0.04 | |
| Density [kg/m$^3$] | 35 | | |
| Specific Heat [J/kgK] | 1464 | | |
| Conductivity [W/mK] | 0.028 | 0.025 | 0.026 |
| Thermal transmittance of 35 cm vertical wall [W/m$^2$K] | 0.383 | 0.359 | 0.367 |
| Thermal transmittance of 60 cm vertical wall [W/m$^2$K] | 0.301 | 0.286 | 0.291 |
| Heating thermal load [kW] | 2.89 | 2.85 | 2.86 |

**Table 5.** Palermo. Thermal characteristics of insulated walls with an EPS layer and heating thermal load.

| Grey EPS | | | |
|---|---|---|---|
| **PALERMO** | **Declared** | **Measured t = 0** | **Measured t = 1** |
| Thickness [m] | 0.04 | | |
| Density [kg/m$^3$] | 10.35 | | |
| Specific Heat [J/kgK] | 1340 | | |
| Conductivity [W/mK] | 0.031 | 0.034 | 0.0341 |
| Thermal transmittance of a 35 cm vertical wall [W/m$^2$K] | 0.404 | 0.424 | 0.424 |
| Thermal transmittance of a 60 cm vertical wall [W/m$^2$K] | 0.314 | 0.325 | 0.326 |
| Heating thermal load [kW] | 2.92 | 2.95 | 2.95 |

**Table 6.** Naples. Thermal characteristics of insulated walls with a polyurethane layer and heating thermal load.

| Polyurethane | | | |
|---|---|---|---|
| **NAPLES** | **Declared** | **Measured t = 0** | **Measured t = 1** |
| Thickness [m] | | 0.06 | |
| Specific Heat [J/kgK] | | 1464 | |
| Density [kg/m$^3$] | | 35 | |
| Conductivity [W/mK] | 0.028 | 0.025 | 0.026 |
| Thermal transmittance of a 35 cm vertical wall [W/m$^2$K] | 0.301 | 0.279 | 0.286 |
| Thermal transmittance of a 60 cm vertical wall [W/m$^2$K] | 0.247 | 0.233 | 0.238 |
| Heating thermal load [kW] | 3.9 | 3.8 | 3.83 |

Supposing a linear performance decay of ETICS, as an example, Figure 4 shows the variation of the thermal load as a function of time for the climatic zone E (the coldest zone).

As can be seen in Figure 5, in the case of polyurethane, there is an increase in thermal transmittance equal to 2% at time T1 compared to time T0. Instead, in the case of grey EPS, this increase is negligible. However, it should be noted that, in the case of EPS, in the wall of 30 cm, although the transmittance value based on the declared data is initially below the limit imposed by the Italian law, this limit is not respected over time. It must be highlighted that the parameter that influences the heating thermal load the most is thermal transmittance, which is closely related to the thickness and conductivity of the insulation materials.

**Table 7.** Naples. Thermal characteristics of insulated walls with an EPS layer and heating thermal load.

| | **Grey EPS** | | |
| --- | --- | --- | --- |
| **NAPLES** | **Declared** | **Measured t = 0** | **Measured t = 1** |
| Thickness [m] | | 0.06 | |
| Specific Heat [J/kgK] | | 1340 | |
| Density [kg/m$^3$] | | 10.35 | |
| Conductivity [W/mK] | 0.031 | 0.034 | 0.0341 |
| Thermal transmittance of a 35 cm vertical wall [W/m$^2$K] | 0.32 | 0.34 | 0.341 |
| Thermal transmittance of a 60 cm vertical wall [W/m$^2$K] | 0.261 | 0.273 | 0.273 |
| Heating thermal load [kW] | 3.9 | 4.0 | 4.12 |

**Table 8.** Turin. Thermal characteristics of insulated wall with a polyurethane layer and heating thermal load.

| | **Polyurethane** | | |
| --- | --- | --- | --- |
| **TURIN** | **Declared** | **Measured t = 0** | **Measured t = 1** |
| Thickness [m] | | 0.1 | |
| Specific Heat [J/kgK] | | 1464 | |
| Density [kg/m$^3$] | | 35 | |
| Conductivity [W/mK] | 0.028 | 0.025 | 0.026 |
| Thermal transmittance of a 35 cm vertical wall [W/m$^2$K] | 0.210 | 0.193 | 0.199 |
| Thermal transmittance of a 60 cm vertical wall [W/m$^2$K] | 0.189 | 0.170 | 0.174 |
| Heating thermal load [kW] | 5.36 | 5.30 | 5.32 |

**Table 9.** Turin. Thermal characteristics of insulated wall with an EPS layer and heating thermal load.

| | **Grey EPS** | | |
| --- | --- | --- | --- |
| **TURIN** | **Declared** | **Measured t = 0** | **Measured t = 1** |
| Thickness [m] | | 0.1 | |
| Specific Heat [J/kgK] | | 1340 | |
| Density [kg/m$^3$] | | 10.35 | |
| Conductivity [W/mK] | 0.031 | 0.0340 | 0.0341 |
| Thermal transmittance of a 35 cm vertical wall [W/m$^2$K] | 0.227 | 0.242 | 0.243 |
| Thermal transmittance of a 60 cm vertical wall [W/m$^2$K] | 0.195 | 0.207 | 0.207 |
| Heating thermal load [kW] | 5.77 | 5.48 | 5.48 |

In this way, when the thermal conductivity of the insulating materials changes over time, even slightly, the thermal load consequently changes.

Similar results occur in the case of Naples (Figure 6) and Turin (Figure 7), where the decay of the insulation performance after 8 years is felt more in the case of polyurethane, increasing the transmittance by about 2–3% both in mild climate and in continental climate. However, as can be seen from the comparisons in terms of transmittance, EPS shows a greater resistance to aging over time compared to polyurethane even if the starter insulation panels are different from the declared one: polyurethane shows a lower thermal conductivity than the declared one, while EPS shows a higher conductivity than the declared one. EPS shows better ageing resistance than polyurethane and this probably depends on the fact that the closed cells structure of EPS ensures that absorption by capillarity is practically nil, especially with reference to the absorption of humid air. The top layers are the same for the two materials.

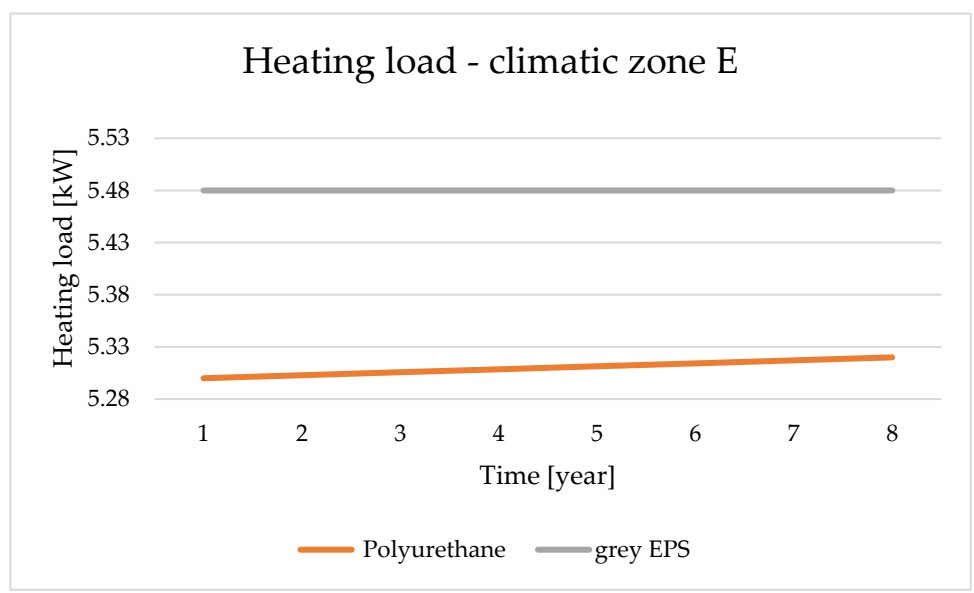

**Figure 4.** Building heating load as a function of time for climatic zone E.

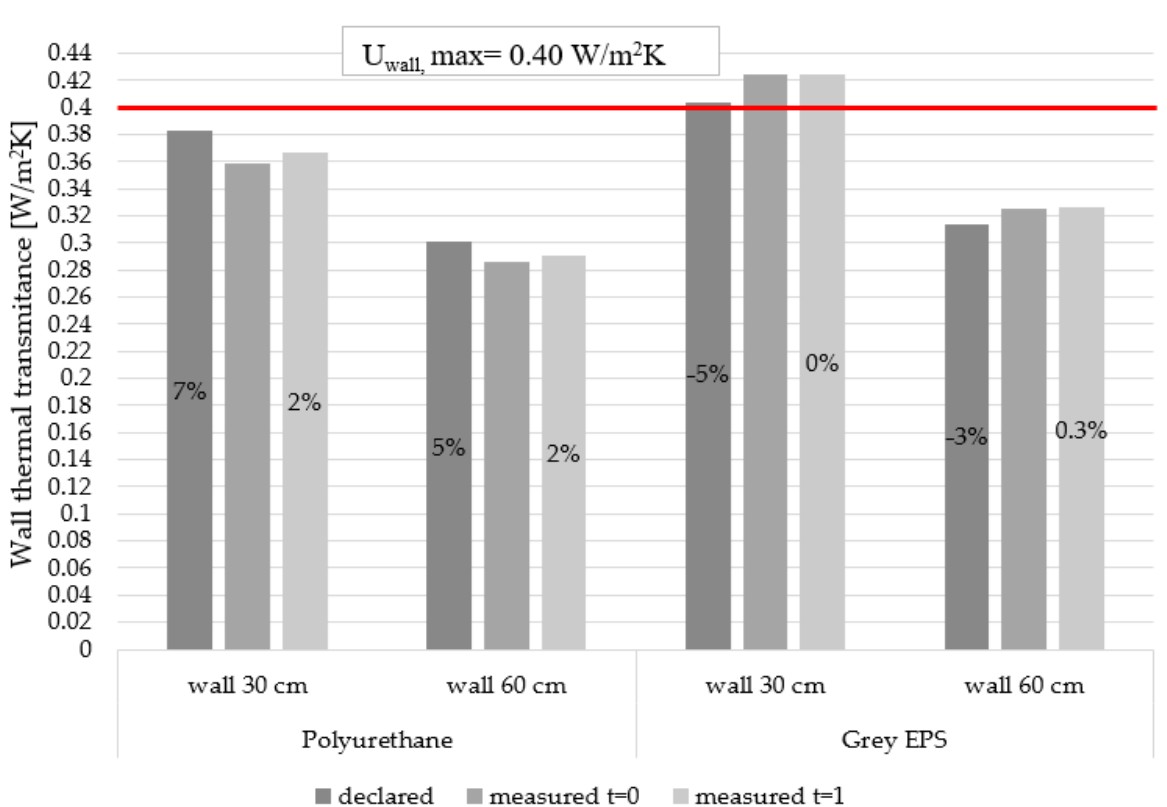

**Figure 5.** Thermal transmittance comparisons for Palermo in the cases of polyurethane and grey EPS.

A further comparison was conducted on the influence of the decay of the performance for the two types of insulation with reference to consumption for heating. As can be seen in Figure 8, the consumption for heating increases only slightly over time, and the greatest percentage of increase occurs in the case of polyurethane. In this case, in hot and mild climates, the increasing percentage is about 2%. On the contrary, the excellent performance in terms of the durability of the EPS led to a negligible variation in energy consumption. By contrast, the consumption for heating, in the case of polyurethane, is always lower than the

expected one based on the declared conductivity, because of the measured conductivities being lower than the declared ones.

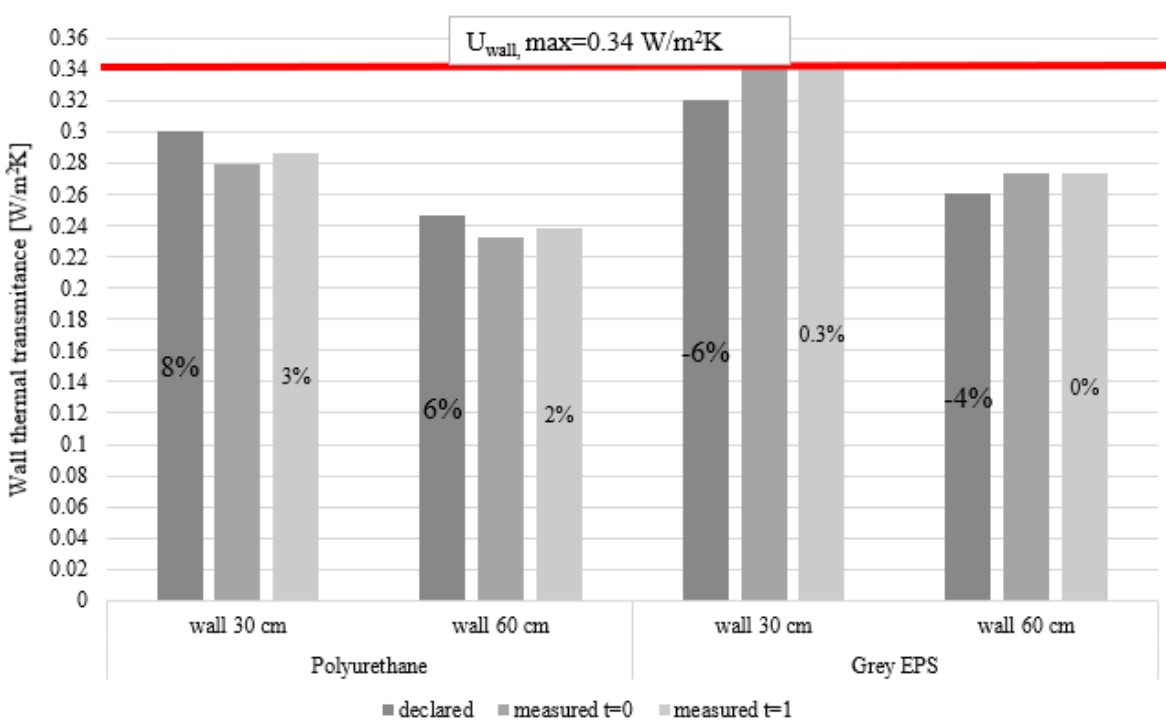

**Figure 6.** Thermal transmittance comparisons for Naples in the cases of polyurethane and grey.

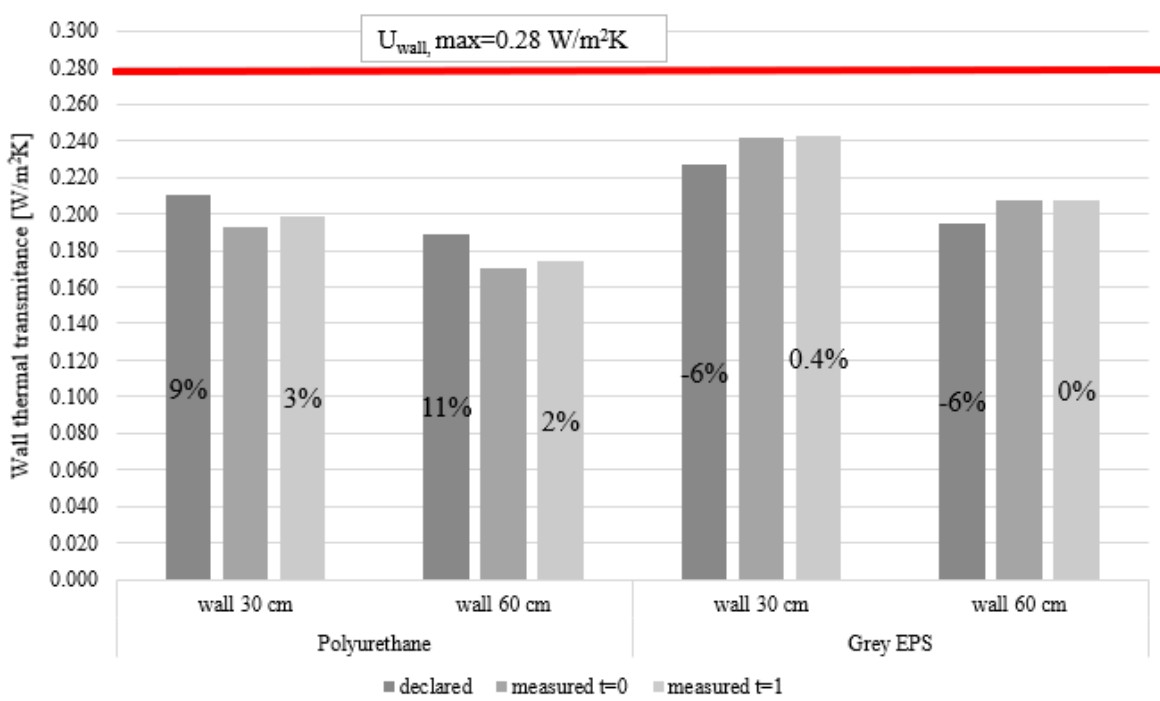

**Figure 7.** Thermal transmittance comparisons for Turin in the cases of polyurethane and grey.

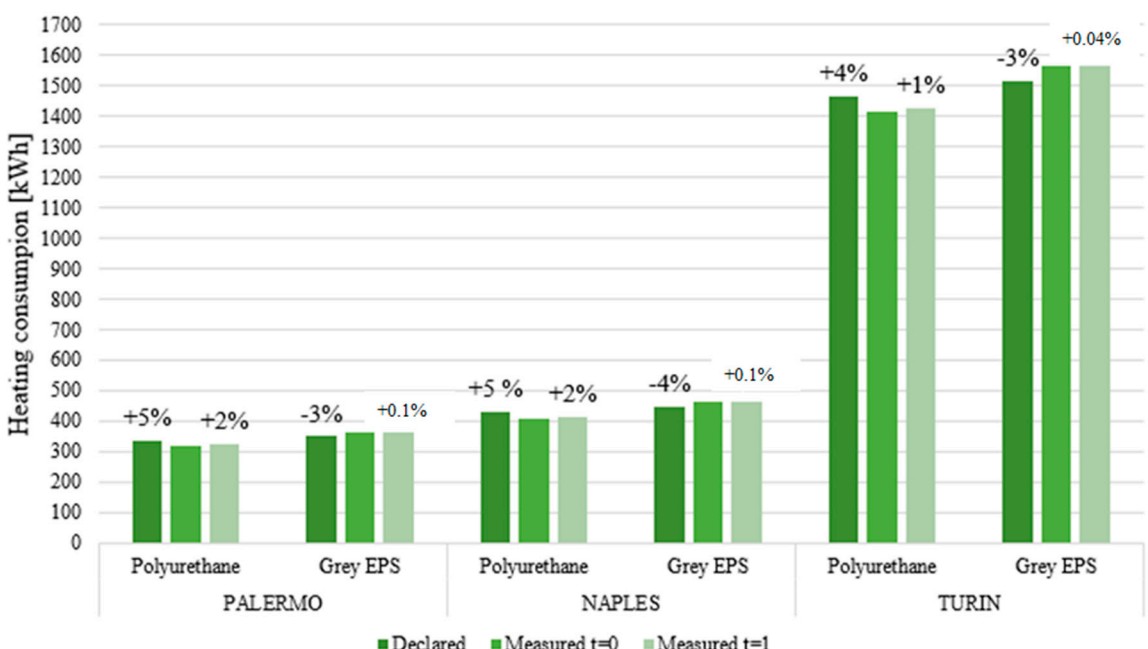

**Figure 8.** Variation in heating energy consumption for different climatic conditions and insulation materials over time.

The results found regarding polyurethane in this research are similar to those obtained by other studies [43,44]. However, it should be noted that the results are somewhat controversial because the increasing conductivity at time T1 is compared to that measured at time T0, while, if compared to the declared conductivity, polyurethane showed decreasing conductivity. As found in several studies [47,48], ETICSs are one of the most relevant energy efficiency interventions on buildings, which allow for significant energy savings for air conditioning (mainly in winter) and reduction in $CO_2$ emissions. The accelerated ageing cycles analyzed in this paper showed that ETICS performances have minor variations over the time. The study demonstrates that, although polyurethane shows a slight increase in thermal conductivity over time, it can be said that, for both the materials analyzed, their decay in performance has little influence on the consumptions related to the building heating.

## 4. Conclusions

In this paper, an experimental study on the decay of two insulating materials was conducted. The measurements show a very slight decrease in thermal performances during ageing: in particular, polyurethane shows a percentage increase in conductivity by only 4%, which can be related to the range of tolerance of the measurement methodology, realized according to EN 12667 standard. The increase in conductivity is completely negligible in the case of EPS compared to the measured one at time T0, but it is more significant (9.7%) if compared to the declared conductivity. This worsening may lead to a failure to comply with the transmittance limit required by law.

The application of the experimental results to a case study building virtually located in three different Italian climatic zones shows that there is a worsening of the transmittance by only 2–3% over time in the case of polyurethane, while this increase is negligible in the case of EPS. Similar worsening rates are found when considering the influence of aging of the insulation on the heating energy requirement for the building in winter. Additionally, in this case, with reference to the polyurethane, there is an increase in the yearly heating consumption of only 1–2% for all the three climatic zones. Although this percentage is low, it should be noted that it refers to a period of about 8 years, a period much shorter than the

useful life of a building. The increase in energy consumption for the building heating is negligible in the case of EPS.

The main limitations of this study are the lack of analyses on the aging of the ETICS for a period greater than 8 years and the evaluation of the energy savings also deriving from the summer behavior of the building envelope. For this reason, future research will include the analysis of the aging for a time equal to the useful life of the ETICS and the investigation on the influence of aging in summer conditions, with reference to dynamic thermal transmittance, attenuation factor and phase shift of the walls. Furthermore, the difference between aging for synthetic insulation and natural insulation will be analyzed.

**Author Contributions:** Conceptualization, D.D., R.L., F.M. and M.N.; methodology, D.D., R.L., F.M. and M.N.; software, D.D., R.L., F.M. and M.N.; formal analysis, D.D., R.L., F.M. and M.N.; resources, D.D., R.L., F.M. and M.N.; data curation, D.D., R.L., F.M. and M.N.; writing—original draft preparation, D.D., R.L., F.M. and M.N.; writing—review and editing, D.D., R.L., F.M. and M.N.; supervision, D.D., R.L., F.M. and M.N. All authors have read and agreed to the published version of the manuscript.

**Funding:** This research received no external funding.

**Institutional Review Board Statement:** Not applicable.

**Informed Consent Statement:** Not applicable.

**Data Availability Statement:** Not applicable.

**Conflicts of Interest:** The authors declare no conflict of interest.

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
