# Peer review of "Experimental Study on the Performance Decay of Thermal Insulation and Related Influence on Heating Energy Consumption in Buildings"

_sustainability, doi:10.3390/su14052947_

Round 1
Reviewer 1 Report
The study is very interesting because it evaluates the relationship between the long-term durability of ETICS and the energy efficiency of the system.
The methodology is clear and the conclusions are well supported by the results presented. However, some clarification is considered appropriate:
- lines 199-200. Probabily, the presence of the masonry support in the sample could change the durability of the system: the vapor permeability of the wall could vary and, consequently, the moisture content could change (inside the insulation, when it is damaged, and in the surface between panel and support). This could alter the behavior of the system.
- lines 249-254. Please clarify this part. It is not clear the relationship between accelerated aging evaluated in the reference [22] and the accelerated aging developed in this research which refers to the EAD to get the ETA
Furthermore, it is advisable to update the references with two recently published articles:
- João Luís Parrachaet al, Impact of natural and artificial aging on the properties of multilayer external wall thermal insulation systems, in Construction and Building Materials, Volume 317, 24 January 2022,
- Joao L. Parracha et al., Effects of hygrothermal, UV and SO2 accelerated ageing on the durability of ETICS in urban environments, in Building and Environment 204 (2021)
Author Response
Dear Reviewers,
in attachment the modified paper considering the Reviewers' comments. In a separate file there are the detailed answers.
We hope that the paper can be accepted in its current form.
Thank You.

Reviewer 2 Report
The manuscript titled "Experimental study on performance decay of thermal insulation and related influence on heating energy consumption in buildings" experimentally studies the degradation of insulation over time and finds a minor increase in building heating requirement (~2% at 8 years). In my opinion, the work is scientifically sound and I would recommend that it be reconsidered after the changes noted below.
- The use of paragraph breaks needs to be improved. Currently, every sentence in abstract and introduction has a paragraph break which makes it hard to follow change in ideas.
- Line 174, section numbering seems to be off.
- Line 147-183 are introduction while Lines 119-144 contain methods for some reason. The organization of thoughts needs to be improved.
- The main point of the paper is to show heating load change as time passes, it would be important to show this data clearly with x-axis of time.
- Why is transmittance different for the different locations? How sensitive are the results to this transmittance?
- What are the most sensitive parameters that affect heating loading over time from this analysis?
Author Response
Dear Reviewers,
in attachment the modified paper considering the Reviewers' comments. In a separate filed you can find the detailed answers.
We hope that the paper can be accepted in its current form.
Thank You.

Reviewer 3 Report
This article is an experimental study of the ageing effects on thermal properties of expanded polystyrene and polyurethane thermal insulation. Overall, the article offers interesting insight about the performance decay of thermal insulation and could be a valuable addition to the literature. At moments the article appears more as a technical report than as a research article. In this regard the following points are to be addressed in order to improve the article:
- The introduction should be more focused on a detailed literature overview about past research on decay and ageing of thermal insulation for buildings. For the major part, the Introduction reports general facts about building energy consumption and thermal insulation. Lines 103-118, which are focused on the subject of research, should be expanded by discussing the research approaches, the major findings, conclusions and limitations of previous studies on thermal insulation decay.
- As consequence of the general Introduction, the scientific contribution of the present study cannot be exactly understood. How do your study compare against previous research? What is new about your experimental approach for the testing of thermal insulation decay? Lines 128-144 suggest that your study is the first on this subject, but later you say that your experimental approach expands previous works (lines 174-178)?
- What is the stance of Italian authorities regarding fire resistance properties of polyurethane and expanded polystyrene? For instance, mineral wool and stone wool perform better in fires and are increasingly recommended for the use in residential buildings, especially high-rise buildings.
- The rendering system (base coat, glass fiber and finishing coat) is the layer that covers and protects the thermal insulation from the adverse external weather conditions? This means that the thermal insulation decay is very much dependent on the quality and durability of this covering layer? Have you come across any research about the ageing properties of the top cover layer?
- There are two Figures 2.
- Line 358: the thermal conductivity of polyurethane increases in the first years of operation as consequence of the diffusion of air and the escape of inert gases - argon from the insulation? This is also because air contains humidity and the penetration of moisture inside the thermal insulation increases heat conduction?
- EPS shows better ageing resistance than PH since the value after accelerated ageing is equal to the initial value λ1 = λ0 = 0.034 W/mK. This means that EPS is not absorbing moisture from humid air? The top layer covers are the same for EPS and PU?
- What procedure is used to obtain the declared value λD, is it measured by the manufacturer, in what way? What is the measurement uncertainties for the data given in Table 3?
- The case study (section 3) is performed in the laboratory climatic chamber or the EPS/PU thermal insulation were installed onto existing buildings in the three Italian cities (lines 392-393)?
- The internal (hi) and external (he) heat transfer coefficients in equation (1) were assumed or calculated? This is important since the values of thermal transmittance (U) and thermal resistance (R) depend on them.
- How did you calculate the heating thermal load (kW) in tables 4-9?
- All the results presented in sections 3 and 4 are actually derived from the measurement data given in Table 3. My suggestion is to strengthen the discussion around Table 3 with measurement uncertainties and quality control of testing procedure. How did you ensure that the measurements are reliable, did you consider performing ageing tests on multiple EPS and PU samples in order to reduce measurement errors (type A and type B uncertainty)?
- Line 481: I suggest caution with a statement that EPS is excellent in terms of durability and that achieves negligible variation of energy consumption. The differences between EPS and PU could be within the measurement uncertainty or caused by errors in testing procedure?
- The estimated period between T0 and T1 is 8 years. This period could be increased to 20 or 30 years by increasing the number of heat/rain, heat/cold and freeze/thaw cycles (table 2) and the ageing effect would be greater?
- The Conclusion should include guidelines for future research, what are the main limitations of the present study and how to overcome them in the future?
Author Response

(The authors gave the same response as above.)

Reviewer 4 Report
The papers deals with Experimental study on performance decay of thermal insulation and related influence on heating energy consumption in buildings. The presentation of a paper and the COMPLETE information ensure its publication. In this case, especially the presentation /shape of the paper must be drastically improved. Text has also must be improved as content. Please see below my suggestions:
Abstract should be reshaped as a single/unique paragraph.
Introduction
- Paragraphs describing the aim of the study - i.e. (L78-79)(L119-144) are mixed with others paragraphs in the Introduction section. Please make a single last paragraph all things developing the aim of the study.
- Moreover, this Introduction is much too long. A good part of it must be moved to Discussion section.
- Many statements are not supported by references. As this suppose to be the literature background, please check and update the literature supporting it.
L176.Consecutive than more 2 references should be written as [18, 19, 27-29]. Please see the instructions for authors in this regard.
Table 1. Please enlarge it on the entire width of the page. Numbers should be written English style (with point not with comma) in the last column.
During the manuscript, there are many sentences considered as separate paragraph. Please merge some of them. The text must be more compact, a paragraph is an idea not a sentence. The text will be easier to follow.
Table 2. Enlarge/extend it on the entire width of the page as the text to be written in a single line not in 3 (i.e.
|
Freeze and thaw |
L328-335, L342-347 - please bullet them; Actual shape is confusing.
Table 3. Please numerical values in English style. Add a 2nd column explaining the thermal characteristics, 1st column will be Symbol, legend under the table will be removed. Much easier to check them
Tables 4, 7, 8, 9. Be consistent with denotation. Replace
|
[kg/m3] |
and
|
[W/m2K] |
with kg/m3 W/m2K. PLease revise and correct the entire manuscript in this regard.
Section 3 presents Results. I suggest naming it Results on a case study (or maybe the authors have a better idea). Moreover, all text in section 4 belongs also to this section, as it presents graphical part.
A new section 5. Discussion section must be developed. Actual Discussion are almost missing. Please compare your obtained results with literature data, maybe in a Table having as the last column References - in this regard I suggest also checking and refer to the following 2 papers: https://doi.org/10.1016/j.scitotenv.2020.137446 ; https://doi.org/10.3390/su11236824 . At the final of Discussion, please highlight in a separate paragraph the strengths and the weakness (if there is one) of your study.
5. Conclusions. They are too long and repetitive with other parts of the manuscript. No need to remember again the aim of the study or numerical values. This part usually is a single paragraph, briefly describing the main achievements of the study. Part of actual Conclusions section can be moved in the future discussion part.
Do not insert references in Conclusions. They are conclusions to your work!
Author Response

(The authors gave the same response as above.)

Round 2
Reviewer 3 Report
The Authors have provided detailed responses to all my comments and improved the article accordingly. I have no further suggestions.
Reviewer 4 Report
The authors responded to my requests.